# Bayesian Discrepancy Measure: Higher-Order and Skewed Approximations

**DOI:** 10.3390/e27070657

**Published:** 2025-06-20

**Authors:** Elena Bortolato, Francesco Bertolino, Monica Musio, Laura Ventura

**Affiliations:** 1Barcelona School of Economics, Universitat Pompeu Fabra, 08005 Barcelona, Spain; elena.bortolato@bse.eu; 2Department of Mathematics and Computer Science, University of Cagliari, 09124 Cagliari, Italy; bertolin@unica.it (F.B.); mmusio@unica.it (M.M.); 3Department of Statistical Sciences, University of Padova, 35121 Padova, Italy

**Keywords:** Bayesian discrepancy measure, credible regions, evidence, higher-order asymptotics, matching priors, nuisance parameter, optimal transport map, precise null hypothesis, skewed approximations, skew-normal distribution, tail area probability

## Abstract

The aim of this paper is to discuss both higher-order asymptotic expansions and skewed approximations for the Bayesian discrepancy measure used in testing precise statistical hypotheses. In particular, we derive results on third-order asymptotic approximations and skewed approximations for univariate posterior distributions, including cases with nuisance parameters, demonstrating improved accuracy in capturing posterior shape with little additional computational cost over simple first-order approximations. For third-order approximations, connections to frequentist inference via matching priors are highlighted. Moreover, the definition of the Bayesian discrepancy measure and the proposed methodology are extended to the multivariate setting, employing tractable skew-normal posterior approximations obtained via derivative matching at the mode. Accurate multivariate approximations for the Bayesian discrepancy measure are then derived by defining credible regions based on an optimal transport map that transforms the skew-normal approximation to a standard multivariate normal distribution. The performance and practical benefits of these higher-order and skewed approximations are illustrated through two examples.

## 1. Introduction

Bayesian inference often relies on asymptotic arguments, leading to approximate methods that frequently assume a parametric form for the posterior distribution. In particular, a Gaussian distribution provides a convenient density for a first-order approximation. This practice is formally justified under regularity conditions by the Bernstein–von Mises theorem. However, this approximation fails to capture potential skewness and asymmetry in the posterior distribution. To avoid this drawback, starting from third-order expansions of Laplace’s method for the posterior distributions (see, e.g., [1,2,3], and references therein), possible alternatives are as follows:Higher-order asymptotic approximations: These offer improved accuracy at minimal additional computational costs compared to first-order approximations, and are applicable to posterior distributions and quantities of interest such as tail probabilities and credible regions (see, e.g., [4], and references therein);To use skewed approximations for the posterior distribution, theoretically justified by a skewed Bernstein–von Mises theorem (see, e.g., [5,6], and references therein).

The aim of this contribution is to discuss higher-order expansions and skew-symmetric approximations for the Bayesian discrepancy measure (BDM) proposed in [7] for testing precise statistical hypotheses. Specifically, the BDM assesses the compatibility of a given hypothesis with the available information (prior and data). To summarize this information, the posterior median is used, providing a straightforward evaluation of the discrepancy with the null hypothesis. The BDM possesses desirable properties such as consistency and invariance under reparameterization, making it a robust measure of evidence.

For a scalar parameter of interest, even with nuisance parameters, computing the BDM involves evaluating tail areas of the posterior or marginal posterior distribution. A first-order Gaussian approximation can be used, but it may be inaccurate, especially with small sample sizes or many nuisance parameters, since it fails to account for potential posterior asymmetry and skewness. In this respect, the aim of this paper is to provide higher-order asymptotic approximations and skewed asymptotic approximations for the BDM. For the third-order approximations, connections with frequentist inference are highlighted when using objective matching priors.

Also, for multidimensional parameters, while a first-order Gaussian approximation of the posterior distribution can be used to calculate the BDM, it still fails to account for potential posterior asymmetry and skewness. In this respect, this paper also addresses higher-order asymptotic approximations and skewed approximations for the BDM. The latter ones are based on an optimal transport map (see [8,9]), which transforms the skew-normal approximation to a standard multivariate normal distribution.

This paper is organized as follows. Section 2 provides some background for the BDM for a scalar parameter of interest, even with nuisance parameters, and extends the definition to the multivariate framework. Section 3 illustrates higher-order Bayesian approximations for the BDM; connections with frequentist inference are highlighted when using objective matching priors. Section 4 discusses skewed approximations for the posterior distribution and for the BDM, theoretically justified by a skewed Bernstein–von Mises theorem, with new insights into the multivariate framework. Two examples are discussed in Section 5. Finally, some concluding remarks are given in Section 6.

## 2. Background

Consider a sampling model f(y;θ), indexed by a parameter $\theta \in \Theta \subseteq R^{d}$, d≥1, and let L(θ)=L(θ;y)=exp{ℓ(θ)} be the likelihood function based on a random sample $y=(y_{1},\ldots ,y_{n})$ of size *n*. Given a prior density π(θ) for θ, Bayesian inference for θ is based on the posterior density π(θ|y)∝π(θ)L(θ).

In several applications, it is of interest to test the precise (or sharp) null hypothesis(1)$$\;H_{0}:\theta =\theta_{0}$$
against $H_{1}:\theta \neq \theta_{0}$. In Bayesian hypothesis testing, the usual approach relies on the well-known Bayes factor (BF), which measures the ratio of posterior to prior odds in favor of the null hypothesis $H_{0}$. Typically, a high BF, or the weight of evidence W=log(BF), provides support for $H_{0}$. However, improper priors can lead to an undetermined BF, and in the context of precise null hypotheses, the BF can be subject to the Jeffreys–Lindley paradox. This paradox highlights a critical divergence between frequentist and Bayesian approaches, that is, as the sample size increases, a *p*-value can become arbitrarily small, leading to the rejection of the null hypothesis, while the BF can simultaneously provide overwhelming evidence in favor of the same precise null. This typically occurs when the alternative hypothesis is associated with a diffuse prior distribution for the parameter of interest. With such priors, the BF tends to favor the simpler $H_{0}$ because, while data might be unlikely under $H_{0}$, it may also be poorly supported by any specific value within the diffuse alternative space. Furthermore, the BF is not well-calibrated, as its finite sampling distribution is generally unknown and may depend on nuisance parameters. To address these limitations, recent research has explored alternative Bayesian measures of evidence for precise null hypothesis testing, including the *e*-value (see, e.g., [10,11,12] and references therein) and the BDM [7]. In the following, we focus on the Bayesian discrepancy measure of evidence proposed in [7] (see also [13]).

### 2.1. Scalar Case

The BDM gives an absolute evaluation of a hypothesis, $H_{0}$, in light of prior knowledge about the parameter and observed data. In the absolutely continuous case, for testing (Equation 1), the BDM is defined as(2)$$\begin{array}{c} \delta_{H}=1-2\;\min\left\{\int_{-\infty }^{\theta_{0}}\pi \left(\theta |y\right)\;d\theta ,1-\int_{-\infty }^{\theta_{0}}\pi \left(\theta |y\right)\;d\theta \right\}. \end{array}$$The quantity $\min\{\int_{-\infty }^{\theta_{0}}\pi \left(\theta |y\right)\;d\theta ,1-\int_{-\infty }^{\theta_{0}}\pi \left(\theta |y\right)\;d\theta \}$ can be interpreted as the posterior probability of a “tail” event concerning only the precise hypothesis $H_{0}$. Doubling this “tail” probability, related to the precise hypothesis $H_{0}$, one obtains a posterior probability assessment about how “central” hypothesis $H_{0}$ is, and, hence, how it is supported by the prior and the data. This interpretation is related to an alternative definition for $\delta_{H}$. Let $\theta_{m}$ be the posterior median and consider the interval defined as $I_{E}=\left(\theta_{0},+\infty \right)$ if $\theta_{m}<\theta_{0}$ or as $I_{E}=\left(-\infty ,\theta_{0}\right)$ if $\theta_{0}<\theta_{m}$. Then, the BDM of the hypothesis $H_{0}$ can be computed as(3)$$\begin{array}{c} \delta_{H}=1-2\;P\left(\theta \in I_{E}|y\right)=1-2\int_{I_{E}}\pi \left(\theta |y\right)\;d\theta . \end{array}$$Note that the quantity $2P(\theta \in I_{E}|y)$ represents the posterior probability of an equi-tailed credible interval for θ.

The Bayesian discrepancy test assesses hypothesis $H_{0}$ based on the BDM. High values of $\delta_{H}$ indicate strong evidence against $H_{0}$, whereas low values suggest data consistency with $H_{0}$. Under $H_{0}$, for large sample sizes, $\delta_{H}$ is asymptotically uniformly distributed on [0,1]. Conversely, when $H_{0}$ is false, $\delta_{H}$ tends to 1 in probability. While thresholds can be set to interpret $\delta_{H}$, in line with the ASA statement, we agree with Fisher that significance levels should be tailored to each case based on evidence and ideas.

The BDM remains invariant under invertible monotonic reparametrizations. Under general regularity conditions and assuming Cromwell’s rule for prior selection, $\delta_{H}$ exhibits specific properties, as follows: (1) if $\theta_{0}=\theta_{t}$ (the true value of the parameter), $\delta_{H}$ tends toward a uniform distribution as the sample size increases; (2) if $\theta_{0}\neq \theta_{t}$, $\delta_{H}$ converges to 1 in probability. Furthermore, using a matching prior, $\delta_{H}$ is exactly uniformly distributed for all sample sizes.

The practical computation of $\delta_{H}$ requires the evaluation of the tail areas of the following form:(4)$$P\left(\theta \geq \theta_{0}|y\right)=\int_{\theta_{0}}^{\infty }\pi \left(\theta |y\right)\;d\theta .$$The derivation of a first-order tail area approximation is simple since it uses a Gaussian approximation. With this approximation, a first-order approximation for $\delta_{H}$ when testing (Equation 1) is simply given by(5)$$\begin{array}{c} \delta_{H}\;\;\dot{=}\;\;2\;\;\Phi \left(\left|\frac{\theta_{0}-\hat{\theta }}{\sqrt{j{\left(\hat{\theta }\right)}^{-1}}}\right|\right)-1, \end{array}$$
where $\hat{\theta }$ is the maximum likelihood estimate (MLE) of θ, $j\left(\theta \right)=-\ell^{\left(2\right)}\left(\theta \right)=-\partial^{2}\ell \left(\theta \right)/\partial \theta^{2}$ is the observed information, the symbol “$\dot{=}$” indicates that the approximation is accurate to $O(n^{-1/2})$, and Φ(·) is the standard normal distribution function. Thus, to first order, $\delta_{H}$ agrees numerically with the 1−p-value based on the Wald statistic $w\left(\theta \right)=\left(\hat{\theta }-\theta \right)/j{\left(\hat{\theta }\right)}^{-1/2}$ and also with the first-order approximation of the *e*-value (see, e.g., [14]). In practice, the approximation (Equation 5) of $\delta_{H}$ may be inaccurate, particularly for a small sample size, because it forces the posterior distribution to be symmetric.

### 2.2. Nuisance Parameters

In most applications, θ is partitioned as θ=(ψ,λ), where ψ is a scalar parameter of interest and λ is a (d−1)—dimensional nuisance parameter, and it is of interest to test the precise (or sharp) null hypothesis(6)$$H_{0}:\psi =\psi_{0}$$
against $H_{1}:\psi \neq \psi_{0}$. In the absolutely continuous case, for testing (Equation 6) in the presence of nuisance parameters, the BDM is defined as(7)$$\begin{array}{c} \delta_{H}=1-2\min\left\{\int_{-\infty }^{\psi_{0}}\pi_{m}\left(\psi |y\right)\;d\psi ,1-\int_{-\infty }^{\psi_{0}}\pi_{m}\left(\psi |y\right)\;d\psi \right\}, \end{array}$$
where $\pi_{m}\left(\psi |y\right)$ is the marginal posterior density for ψ, given by(8)$$\begin{array}{c} \pi_{m}\left(\psi |y\right)=\int \pi \left(\psi ,\lambda |y\right)\;d\lambda \propto \int \pi \left(\psi ,\lambda \right)L\left(\psi ,\lambda \right)\;d\lambda . \end{array}$$

Also in this framework, the practical computation of $\delta_{H}$ requires the evaluation of tail areas of the following form:(9)$$P_{m}\left(\psi \geq \psi_{0}|y\right)=\int_{\psi_{0}}^{\infty }\pi_{m}\left(\psi |y\right)\;d\psi .$$The derivation of a first-order tail area approximation is still simple since it uses a Gaussian approximation. Let $\ell_{p}\left(\psi \right)=\log L\left(\psi ,{\hat{\lambda }}_{\psi }\right)$ be the profile log-likelihood for ψ, where ${\hat{\lambda }}_{\psi }$ denotes the constrained MLE of λ given ψ. Moreover, let $\left(\hat{\psi },\hat{\lambda }\right)$ be the full MLE, and let $j_{p}\left(\psi \right)=-\ell_{p}^{\left(2\right)}\left(\psi \right)=-\partial^{2}\ell_{p}\left(\psi \right)/\partial \psi^{2}$ be the profile observed information. A first-order approximation for $\delta_{H}$ when testing (Equation 6) is simply given by(10)$$\begin{array}{c} \delta_{H}\;\;\dot{=}\;\;2\;\;\Phi \left(\left|\frac{\psi_{0}-\hat{\psi }}{\sqrt{j_{p}{\left(\hat{\psi }\right)}^{-1}}}\right|\right)-1. \end{array}$$Thus, to first order, $\delta_{H}$ agrees numerically with 1−p-value based on the profile Wald statistic $w_{p}\left(\psi \right)=\left(\hat{\psi }-\psi \right)/j_{p}{\left(\hat{\psi }\right)}^{-1/2}$. In practice, as in the scalar parameter case, the approximation (Equation 5) of $\delta_{H}$ may be inaccurate, particularly for a small sample size or a large number of nuisance parameters, since it fails to account for potential posterior asymmetry and skewness.

### 2.3. The Multivariate Case

Extending the definition of the BDM to the multivariate setting, where $\theta \in \Theta \subseteq R^{d}$ with d>1, presents some challenges. The core concepts of the univariate definition rely on the unique ordering of the real line and the uniquely defined median, which splits the probability mass into two equal halves (tail areas). In $R^{d}$, with d>1, there is no natural unique ordering, and concepts like the median and “tail areas” relative to a specific point $\theta_{0}$ lack a single, universally accepted definition. Despite these challenges, the fundamental goal remains the same, that is, to quantify how consistent the hypothesized value $\theta_{0}$ is with the posterior distribution π(θ|y); specifically measuring how “central” or, conversely, how “extreme” $\theta_{0}$ lies within the posterior distribution.

Utilizing the notion of center-outward quantile functions ([8,9]), a concept from recent multivariate statistics, provides a theoretically appealing way to define the multivariate BDM. Let $F_{P}^{\pm }:R^{d}\to B_{d}$ be the *center-outward distribution function* mapping the posterior distribution $P_{\theta }$ (with density π(θ|y)) to the uniform distribution $U_{d}$ on the unit ball $B_{d}$. More precisely, the *center-outward distribution function* $F_{P}^{\pm }:R^{d}\to B_{d}$ is defined as the *almost-everywhere unique gradient of a convex function* that pushes a distribution $P_{\theta }$ forward to the uniform distribution $U_{d}$ on the unit ball $B_{d}$ in $R^{d}$. That is,$$F_{P}^{\pm }:=\nabla g,\;\mathrm{such}\;\mathrm{that}\;F_{P}^{\pm }\#P_{\theta }=U_{d}.$$The *center-outward quantile function* $Q_{P}^{\pm }$ is defined as the (continuous) inverse of $F_{P}^{\pm }$, i.e.,$$Q_{P}^{\pm }:={\left(F_{P}^{\pm }\right)}^{-1}.$$It maps the open unit ball $B_{d}$ (minus the origin) to $R^{d}\setminus {\left(F_{P}^{\pm }\right)}^{-1}\left(0\right)$ and satisfies$$Q_{P}^{\pm }\#U_{d}=P_{\theta }.$$For τ∈(0,1), we define the *center-outward quantile region of order τ* as$$R_{P}^{\pm }\left(\tau \right):=Q_{P}^{\pm }\left(\tau \;B_{d}\right),$$
and the *center-outward quantile contour of order τ* as$$C_{P}^{\pm }\left(\tau \right):=Q_{P}^{\pm }\left(\tau \;S^{d-1}\right),$$
where $S^{d-1}$ is the unit sphere in $R^{d}$. When d=1, this coincides with the rescaled univariate cumulative distribution function $F_{P}^{\pm }\left(x\right)=2F_{P}\left(x\right)-1$ and the BDM (Equation 7) can be expressed as$$\delta_{H}=\left|F_{P}^{\pm }\left(\theta_{0}\right)\right|.$$This measures the (rescaled) distance of the quantile rank of $\theta_{0}$ from the center point (corresponding to rank 0). Generalizing this, we can define the multivariate BDM for the hypothesis $H_{0}:\theta =\theta_{0}$ as(11)$$\delta_{H}=∥F_{P}^{\pm }\left(\theta_{0}\right)∥,$$
where ∥·∥ denotes the standard Euclidean norm in $R^{d}$. Here, $F_{P}^{\pm }\left(\theta_{0}\right)$ maps the point $\theta_{0}$ to a location u within the unit ball $B_{d}$. This definition has desirable properties (see [8]):It yields a value between 0 and 1;$\delta_{H}=0$ if $\theta_{0}$ corresponds to the geometric center (or multivariate median) of the distribution (mapped to 0 by $F_{P}^{\pm }$);$\delta_{H}$ increases as $\theta_{0}$ moves away from the center toward the “boundary” of the distribution, approaching 1 for points mapped near the surface of the unit ball $S^{d-1}$;It is invariant under suitable classes of transformations (affine transformations if $P_{\theta }$ is elliptically contoured, more generally under monotone transformations linked to an optimal transport map construction);It naturally reduces to the univariate definition $\delta_{H}=\left|F_{P}^{\pm }\left(\theta_{0}\right)\right|$ when d=1.

The primary practical difficulty lies in computing the center-outward distribution function $F_{P}^{\pm }\left(\cdot \right)$ for an arbitrary posterior distribution π(θ|y), as it typically requires solving a complex optimal transport problem (see [15]).

## 3. Beyond Gaussian I: Higher-Order Asymptotic Approximations

### 3.1. Scalar Case

In order to have more accurate evaluations of the first-order approximation (Equation 5) of $\delta_{H}$, it may be useful to resort to higher-order approximations based on tail area approximations (see, e.g., [3,4], and references therein). Using the tail area argument to the posterior density, we can derive the $O(n^{-3/2})$ approximation:(12)$$\begin{array}{c} P\left(\theta \geq \theta_{0}|y\right)\;\;\ddot{=}\;\;\Phi \left(r_{B}^{*}\left(\theta_{0}\right)\right), \end{array}$$
where the symbol “$\ddot{=}$” indicates that the approximation is accurate to $O(n^{-3/2})$ and$$r_{B}^{*}\left(\theta \right)=r\left(\theta \right)+\frac{1}{r(\theta )}\log\frac{q(\theta )}{r(\theta )},$$
with $r\left(\theta \right)=\mathrm{sign}\left(\hat{\theta }-\theta \right){\left[2\left(\ell \left(\hat{\theta }\right)-\ell \left(\theta \right)\right)\right]}^{1/2}$, the likelihood root, and$$q\left(\theta \right)=\ell^{\left(1\right)}\left(\theta \right)\;j{\left(\hat{\theta }\right)}^{-1/2}\frac{\pi (\hat{\theta })}{\pi (\theta )}.$$In the expression of q(θ), $\ell^{\left(1\right)}\left(\theta \right)=\partial \ell \left(\theta \right)/\partial \theta$ is the score function.

Using the tail area approximation (Equation 12), a third-order approximation of the BDM (Equation 2) can be computed as(13)$$\begin{array}{c} \delta_{H}\;\;\ddot{=}\;\;1-2\;\min\left\{\Phi \left(r_{B}^{*}\left(\theta_{0}\right)\right),1-\Phi \left(r_{B}^{*}\left(\theta_{0}\right)\right)\right\}=2\Phi (|r_{B}^{*}\left(\theta_{0}\right)\left|\right)-1. \end{array}$$Note that the higher-order approximation (Equation 13) does not call for any condition on the prior π(θ), i.e., it can also be improper, and it is available at a negligible additional computational cost over the simple first-order approximation.

Note also that using $r_{B}^{*}\left(\theta \right)$ an (1−α) equi-tailed credible interval for θ can be computed as $CI=\{\theta :|r_{B}^{*}\left(\theta \right)|\leq z_{1-\alpha /2}\}$, where $z_{1-\alpha /2}$ is the (1−α/2)-quantile of the standard normal distribution, and in practice, it can reflect asymmetries of the posterior. Moreover, from (Equation 12), the posterior median can be computed as the solution for θ of the estimating equation $r_{B}^{*}\left(\theta \right)=0$.

### 3.2. Nuisance Parameters

When θ is partitioned as θ=(ψ,λ), where ψ is a scalar parameter of interest and λ is a (d−1)—dimensional nuisance parameter, in order to have more accurate evaluations of the first-order approximation (Equation 10) of $\delta_{H}$, using the tail area argument to the marginal posterior density, we can derive the $O(n^{-3/2})$ approximation (see, e.g., [3,4]):(14)$$\begin{array}{c} P_{m}\left(\psi \geq \psi_{0}|y\right)\;\;\ddot{=}\;\;\Phi \left(r_{Bp}^{*}\left(\psi_{0}\right)\right), \end{array}$$
where$$r_{Bp}^{*}\left(\psi \right)=r_{p}\left(\psi \right)+\frac{1}{r_{p}\left(\psi \right)}\log\frac{q_{B}\left(\psi \right)}{r_{p}\left(\psi \right)},$$
with $r_{p}\left(\psi \right)=\mathrm{sign}\left(\hat{\psi }-\psi \right){\left[2\left(\ell_{p}\left(\hat{\psi }\right)-\ell_{p}\left(\psi \right)\right)\right]}^{1/2}$, the profile likelihood root, and$$q_{B}\left(\psi \right)=\ell_{p}^{\left(1\right)}\left(\psi \right){\left|j_{p}\left(\hat{\psi }\right)\right|}^{-1/2}\frac{|j_{\lambda \lambda }\left(\psi ,{\hat{\lambda }}_{\psi }\right){|}^{1/2}}{|j_{\lambda \lambda }\left(\hat{\psi },\hat{\lambda }\right){|}^{1/2}}\frac{\pi (\hat{\psi },\hat{\lambda })}{\pi (\psi ,{\hat{\lambda }}_{\psi })}.$$In the expression of $q_{B}\left(\psi \right)$, $\ell_{p}^{\left(1\right)}\left(\psi \right)$ is the profile score function and $j_{\lambda \lambda }\left(\psi ,\lambda \right)$ represents the (λ,λ)-block of the observed information j(ψ,λ).

Using the tail area approximation (Equation 14), a third-order approximation of the BDM (Equation 7) can be computed as(15)$$\begin{array}{c} \delta_{H}\;\;\ddot{=}\;\;1-2\min\left\{\Phi \left(r_{Bp}^{*}\left(\psi_{0}\right)\right),1-\Phi \left(r_{Bp}^{*}\left(\psi_{0}\right)\right)\right\}=2\Phi (|r_{Bp}^{*}\left(\psi_{0}\right)\left|\right)-1. \end{array}$$Note that the higher-order approximation (Equation 15) does not call for any condition on the prior π(ψ,λ), i.e., it can also be improper. Note also that, using $r_{Bp}^{*}\left(\psi \right)$, a (1−α) equi-tailed credible interval for ψ can be computed as $CI=\{\psi :|r_{Bp}^{*}\left(\psi \right)|\leq z_{1-\alpha /2}\}$. Moreover, from (Equation 14), the posterior median of (Equation 8) can be computed as the solution for ψ of the estimating equation $r_{Bp}^{*}\left(\psi \right)=0$.

#### Approximations with Matching Priors

The order of the approximations in the previous sections refers to the posterior distribution function and may depend, to varying degrees, on the choice of prior. A so-called strong-matching prior (see [16], and references therein) ensures that a frequentist *p*-value coincides with a Bayesian posterior survivor probability to a high degree of approximation, in the marginal posterior density (Equation 8).

Welch and Peers [17] showed that for a scalar parameter θ, Jeffreys’ prior is probability matching, in the sense that posterior survivor probabilities agree with frequentist probabilities, and credible intervals of a chosen width coincide with frequentist confidence intervals. With Jeffreys’ prior, we have$$q\left(\theta \right)=\ell^{\left(1\right)}\left(\theta \right)j{\left(\hat{\theta }\right)}^{-1/2}\frac{i{\left(\hat{\theta }\right)}^{1/2}}{i{\left(\theta \right)}^{1/2}}$$
and the corresponding $r_{B}^{*}\left(\theta \right)$ coincides with the frequentist modified likelihood root as defined by [18]. In this case, using the tail area approximation (Equation 12), a third-order approximation of the BDM of the hypothesis $H_{0}:\theta =\theta_{0}$ coincides with $1-p^{*}$, where $p^{*}$ is the *p*-value based on the frequentist modified likelihood root. Thus, when using Jeffreys’ prior and higher-order asymptotics in the scalar case, there is agreement between Bayesian and frequentist hypothesis testing.

In the presence of nuisance parameters, following [4], when using a strong matching prior, the marginal posterior density can be written as(16)$$\begin{array}{c} \pi_{m}\left(\psi |y\right)\;\;\ddot{\propto }\;\;\exp\left(-\frac{1}{2}r_{p}^{*}{\left(\psi \right)}^{2}\right)\left|\frac{s_{p}\left(\psi \right)}{r_{p}\left(\psi \right)}\right|, \end{array}$$
where $s_{p}\left(\psi \right)=\ell_{p}^{\left(1\right)}\left(\psi \right)/j_{p}{\left(\hat{\psi }\right)}^{1/2}$ is the profile score statistic, and $r_{p}^{*}\left(\psi \right)$ is the modified profile likelihood root:(17)$$\begin{array}{c} r_{p}^{*}\left(\psi \right)=r_{p}\left(\psi \right)+\frac{1}{r_{p}\left(\psi \right)}\log\frac{q_{p}\left(\psi \right)}{r_{p}\left(\psi \right)}, \end{array}$$
which has a third-order standard normal null distribution. In (Equation 17), the quantity $q_{p}\left(\psi \right)$ is a suitably defined correction term (see, e.g., [18] and [19], Chapter 9). Moreover, the tail area of the marginal posterior for ψ can be approximated to third order as(18)$$\begin{array}{c} \;P_{m}\left(\psi \geq \psi_{0}|y\right)\;\;\ddot{=}\;\Phi \left(r_{p}^{*}\left(\psi_{0}\right)\right), \end{array}$$A remarkable advantage of (Equation 16) and (Equation 18) is that their expressions automatically include the matching prior, without requiring its explicit computation.

Using (Equation 18), an asymptotic equi-tailed credible interval for ψ can be computed as $CI=\{\psi :|r_{p}^{*}\left(\psi \right)|\leq z_{1-\alpha /2}\}$, i.e., as a confidence interval for ψ based on (Equation 17) with approximate level (1−α). Note from (Equation 18) that the posterior median of $\pi_{m}\left(\psi |y\right)$ can be computed as the solution for ψ of the estimating equation $r_{p}^{*}\left(\psi \right)=0$, and thus it coincides with the frequentist estimator defined as the zero-level confidence interval based on $r_{p}^{*}\left(\psi \right)$. Such an estimator has been shown to be a refinement of the MLE $\hat{\psi }$.

Using the tail area approximation (Equation 18), a third-order approximation of the BDM of the hypothesis $H_{0}:\psi =\psi_{0}$ is(19)$$\begin{array}{c} \delta_{H}^{*}\ddot{=}1-2\min\left\{\Phi \left(r_{p}^{*}\left(\psi_{0}\right)\right),1-\Phi \left(r_{p}^{*}\left(\psi_{0}\right)\right)\right\}=2\;\Phi (|r_{p}^{*}\left(\psi_{0}\right)\left|\right)-1. \end{array}$$In this case, (Equation 19) coincides with $1-p_{r}^{*}$, where $p_{r}^{*}$ is the *p*-value based on (Equation 17). Thus, when using strong matching priors and higher-order asymptotics, there is agreement between Bayesian and frequentist hypothesis testing, point estimation, and interval estimation.

From a practical point of view, the computation of (Equation 19) can be easily performed in practical problems using the likelihoodAsy package [20] of the statistical software R version 4.4.1. In practice, the advantage of using this package is that it does not require the function $q_{p}\left(\psi \right)$ explicitly, but instead requires only the code for computing the log-likelihood function and for generating data from the assumed model. Some examples can be found in [14].

### 3.3. Multidimensional Parameters

When θ is multidimensional, the derivation of a first-order tail area approximation and a first-order approximation for $\delta_{H}$ remains straightforward, starting from the Laplace approximation of the posterior distribution. In particular, let $W\left(\theta \right)=2(\ell \left(\hat{\theta }\right)-\ell \left(\theta \right))$ be the log-likelihood ratio for θ. Using W(θ), a first-order approximation of the BDM for the hypothesis $H_{0}:\theta =\theta_{0}$ can be obtained as follows:(20)$$\begin{array}{c} \delta_{H}\;\dot{=}\;1-P\left(\chi_{d}^{2}\geq W\left(\theta_{0}\right)\right), \end{array}$$
where $\chi_{d}^{2}$ is the Chi-squared distribution with *d* degrees of freedom. This approximation is asymptotically equivalent to the first-order approximation:(21)$$\begin{array}{c} \delta_{H}\;\dot{=}\;1-P\left(\chi_{d}^{2}\geq {\left(\theta_{0}-\hat{\theta }\right)}^{T}j\left(\hat{\theta }\right)\left(\theta_{0}-\hat{\theta }\right)\right). \end{array}$$

Higher-order approximations based on modifications of the log-likelihood ratios are also available for multidimensional parameters of interest, both with or without nuisance parameters (see [4,19,21], and references therein). As is the case with the approximations for a scalar parameter, the proposed results are based on the asymptotic theory of modified log-likelihood ratios [21], they require only routine maximization output for their implementation, and they are constructed for arbitrary prior distributions. For instance, paralleling the scalar parameter case, a credible region for a *d*-dimensional parameter of interest θ with approximately 100(1−α)% coverage in repeat sampling, can be computed as $CR=\{\theta :W^{*}\left(\theta \right)\leq \chi_{d;1-\alpha }^{2}\}$, where $W^{*}\left(\theta \right)$ is a suitable modification of the log-likelihood ratio W(θ) or of the profile log-likelihood ratio (see [19,21]), and $\chi_{d;1-\alpha }^{2}$ is the (1−α) quantile of the $\chi_{d}^{2}$ distribution. In practice, the CR region can be interpreted as the extension to the multidimensional case of the equi-tailed CI set, i.e., the CR region is computed as a multidimensional case of the CI set based on the Chi-squared approximation. As in the scalar case, the CR region can reflect departures from symmetry with respect to the first-order approximation based on the Wald statistic. Some simulation studies on CR based on $W^{*}\left(\theta \right)$ can be found in [22].

Using $W^{*}\left(\theta \right)$, a higher-order approximation of the BDM for the hypothesis $H_{0}:\theta =\theta_{0}$ can be obtained as(22)$$\begin{array}{c} \delta_{H}\;\ddot{=}\;1-P\left(\chi_{d}^{2}\geq W^{*}\left(\theta_{0}\right)\right). \end{array}$$The major drawback with this approximation is that the signed root log-likelihood ratio transformation $W^{*}\left(\theta \right)$ in general depends on the chosen parameter order. Moreover, its computation can be cumbersome when *d* is large.

## 4. Beyond Gaussian II: Skewed Approximations

A major limitation of standard first-order Gaussian approximations, like (Equation 5) and (Equation 10), is their reliance on symmetric densities, which simplifies inference but can misrepresent key posterior features like skewness and heavy tails. Indeed, even simple parametric models can yield asymmetric posteriors, leading to biased and inaccurate approximations.

To overcome this, recent work has introduced flexible families of approximating posterior densities that can capture the shape and skewness [5,6,23]. In particular, [5] developed a class of closed-form deterministic approximations using a third-order extension of the Laplace approximation. This approach yields tractable, skewed approximations that better capture the actual shape of the target posterior while remaining computationally efficient.

Also, the skewed approximations, as well as the higher-order approximations discussed in Section 3, rely on higher-order expansions and derivatives. They start with a symmetric Gaussian approximation, but centered at the maximum a posteriori (MAP) estimate, and introduce skewness through the Gaussian distribution function combined with a cubic term influenced by the third derivative of the log-likelihood function.

### 4.1. Scalar Case

Let us denote with $\ell^{\left(k\right)}\left(\theta \right)$ the *k*-th derivative of the log-likelihood ℓ(θ), i.e., $\ell^{\left(k\right)}\left(\theta \right)=\partial^{k}\ell \left(\theta \right)/\partial \theta^{k}$, k=1,2,3,…. Moreover, let $\tilde{\theta }={\mathrm{argmax}}_{\theta \in \Theta }\left\{\ell \left(\theta \right)+\log\pi \left(\theta \right)\right\}$ be the MAP estimate of θ and let $h=\sqrt{n}\left(\theta -\tilde{\theta }\right)$ be the rescaled parameter. Using result (14) of [5] and all the regularity conditions there stated, the skew-symmetric (SKS) approximation for the posterior density for θ is(23)$$\;\pi_{SKS}\left(\theta |y\right)\propto 2\;\phi \left(h;0,\tilde{\omega }\right)\;\Phi \left(\tilde{\alpha }\left(h\right)\right),$$
where $\phi (h;0,\tilde{\omega })$ is the normal density function with mean 0 and variance $\tilde{\omega }=nj{\left(\tilde{\theta }\right)}^{-1}$ and$$\tilde{\alpha }\left(h\right)=\frac{\ell^{\left(3\right)}\left(\tilde{\theta }\right)\sqrt{2\pi }}{12n^{3/2}}h^{3}$$
is the skewness component, expressed as a cubic function of *h*, reflecting the influence of the third derivative of the log-likelihood on the shape of the posterior distribution.

Equation (Equation 23) provides a practical skewed second-order approximation of the target posterior density, centered at its mode. This approach is known as the SKS approximation or skew-modal approximation. Compared to the classical first-order Gaussian approximation derived from the Laplace method, the SKS approximation remains similarly tractable while providing significantly greater accuracy. Note that this approximation depends on the prior distribution through the MAP.

Using (Equation 23) and the approximation$$2\;\phi \left(h;0,\tilde{\omega }\right)\left(\frac{1}{2}+\frac{1}{\sqrt{2\pi }}\tilde{\alpha }\left(h\right)\right)=2\;\phi \left(h;0,\tilde{\omega }\right)\Phi \left(\tilde{\alpha }\left(h\right)\right)+O\left(n^{-1}\right),$$
we can derive the approximation$$P_{SKS}\left(\theta \geq \theta_{0}|y\right)=\frac{\int_{h_{0}}^{\infty }2\;\phi \left(h;0,\tilde{\omega }\right)\left(\frac{1}{2}+\frac{1}{\sqrt{2\pi }}\tilde{\alpha }\left(h\right)\right)\;dh}{\int_{-\infty }^{\infty }2\;\phi \left(h;0,\tilde{\omega }\right)\left(\frac{1}{2}+\frac{1}{\sqrt{2\pi }}\tilde{\alpha }\left(h\right)\right)\;dh}$$
for the tail area for (Equation 4), where $h_{0}=\sqrt{n}\left(\theta_{0}-\tilde{\theta }\right)$. Note that the denominator is simply equal to 1, due to the symmetry of ϕ(·) and the oddness of $\tilde{\alpha }\left(h\right)$. The numerator can be split into two integrals:$$\int_{h_{0}}^{\infty }2\phi \left(h;0,\tilde{\omega }\right)\left(\frac{1}{2}+\frac{1}{\sqrt{2\pi }}\tilde{\alpha }\left(h\right)\right)\;dh=\int_{h_{0}}^{\infty }\phi \left(h;0,\tilde{\omega }\right)\;dh+\frac{\sqrt{2}}{\sqrt{\pi }}\int_{h_{0}}^{\infty }\phi \left(h;0,\tilde{\omega }\right)\tilde{\alpha }\left(h\right)\;dh.$$The first integral can be expressed as the standard Gaussian tail, as follows:$$\int_{h_{0}}^{\infty }\phi \left(h;0,\tilde{\omega }\right)\;dh=\left(1-\Phi \left(\frac{h_{0}}{\sqrt{\tilde{\omega }}}\right)\right),$$
while the second integral involves the skewness term and can be expressed as$$\frac{\sqrt{2}}{\sqrt{\pi }}\int_{h_{0}}^{\infty }\phi \left(h;0,\tilde{\omega }\right)\tilde{\alpha }\left(h\right)\;dh=\frac{\ell^{\left(3\right)}\left(\tilde{\theta }\right)}{6n^{3/2}}\int_{h_{0}}^{\infty }h^{3}\phi \left(h;0,\tilde{\omega }\right)\;dh.$$Substituting $z=h/\sqrt{\tilde{\omega }}$ into the integral $\int_{h_{0}}^{\infty }h^{3}\phi \left(h;0,\tilde{\omega }\right)\;dh$, we have$$\begin{array}{cc} \int_{h_{0}}^{\infty }h^{3}\phi \left(h;0,\tilde{\omega }\right)\;dh & =\int_{h_{0}}^{\infty }h^{3}\frac{1}{\sqrt{2\pi \tilde{\omega }}}\exp\left(-\frac{h^{2}}{2\tilde{\omega }}\right)dh\\ \; & =\int_{h_{0}/\sqrt{\tilde{\omega }}}^{\infty }{\left(\sqrt{\tilde{\omega }}z\right)}^{3}\frac{1}{\sqrt{2\pi \tilde{\omega }}}\exp\left(-\frac{{\left(\sqrt{\tilde{\omega }}z\right)}^{2}}{2\tilde{\omega }}\right)\sqrt{\tilde{\omega }}dz\\ \; & ={\tilde{\omega }}^{3/2}\int_{h_{0}/\sqrt{\tilde{\omega }}}^{\infty }z^{3}\frac{1}{\sqrt{2\pi }}\exp\left(-\frac{z^{2}}{2}\right)dz. \end{array}$$Using the identity $\int_{z_{0}}^{\infty }z^{3}\phi \left(z;0,1\right)dz=\phi \left(z_{0};0,1\right)\left(z_{0}^{2}+2\right)$, with $z_{0}=h_{0}/\sqrt{\tilde{\omega }}$, and $\int_{-\infty }^{z_{0}}z^{3}\phi \left(z;0,1\right)dz=-\phi \left(z_{0};0,1\right)\left(z_{0}^{2}+2\right)$, we obtain$$\begin{array}{cc} \int_{h_{0}}^{\infty }h^{3}\phi \left(h;0,\tilde{\omega }\right)\;dh & ={\tilde{\omega }}^{3/2}\phi \left(\frac{h_{0}}{\sqrt{\tilde{\omega }}};0,1\right)\left({\left(\frac{h_{0}}{\sqrt{\tilde{\omega }}}\right)}^{2}+2\right)\\ \; & ={\tilde{\omega }}^{3/2}\phi \left(\frac{h_{0}}{\sqrt{\tilde{\omega }}};0,1\right)\left(\frac{h_{0}^{2}}{\tilde{\omega }}+2\right). \end{array}$$Then, the resulting SKS approximation to $P(\theta \geq \theta_{0}|y)$ is$$\begin{array}{c} P_{SKS}\left(\theta \geq \theta_{0}|y\right)=\left(1-\Phi \left(\frac{h_{0}}{\sqrt{\tilde{\omega }}}\right)\right)+\;\frac{\ell^{\left(3\right)}\left(\tilde{\theta }\right)}{6n^{3/2}}\;{\tilde{\omega }}^{3/2}\phi \left(\frac{h_{0}}{\sqrt{\tilde{\omega }}};0,1\right)\left(\frac{h_{0}^{2}}{\tilde{\omega }}+2\right). \end{array}$$Finally, substituting this approximation into (Equation 7), we obtain the SKS approximation of the BDM, given by(24)$$\begin{array}{c} \delta_{H}^{SKS}=2\;\;\Phi \left(\left|\frac{h_{0}}{\sqrt{\tilde{\omega }}}\right|\right)-2\;\mathrm{sign}\left(h_{0}\right)\;\frac{\ell^{\left(3\right)}\left(\tilde{\theta }\right)}{6n^{3/2}}\;{\tilde{\omega }}^{3/2}\phi \left(\frac{h_{0}}{\sqrt{\tilde{\omega }}};0,1\right)\left(\frac{h_{0}^{2}}{\tilde{\omega }}+2\right)-1. \end{array}$$Note that the first term of this approximation differs from that in (Equation 5) since it is evaluated at the MAP and not at the MLE.

### 4.2. Nuisance Parameters

As in Section 2.2, suppose that the parameter is partitioned as θ=(ψ,λ), where ψ is a scalar parameter of interest and λ is a nuisance parameter of dimension d−1. Also, for the marginal posterior distribution $\pi_{m}\left(\psi |y\right)$, an SKS approximation is available (see [5], Section 4.2).

Adopting the index notation, let us denote by $j\left(\theta \right)=-[\ell_{st}^{\left(2\right)}\left(\theta \right)]$ the observed Fisher information matrix, where $\ell_{st}^{\left(2\right)}\left(\theta \right)=\frac{\partial^{2}\ell \left(\theta \right)}{\partial \theta_{s}\partial \theta_{t}}$, s,t=1,…,d, and let $\Omega ={\left(j\left(\tilde{\theta }\right)/n\right)}^{-1}$ be the inverse of the scaled observed Fisher information matrix evaluated at the MAP. We denote the elements of Ω by $\Omega_{st}$; in particular, let us denote by $\Omega_{11}$ the element corresponding to the parameter of interest, ψ. Moreover, let us denote with $\ell_{stl}^{\left(3\right)}\left(\theta \right)=\frac{\partial^{3}\ell \left(\theta \right)}{\partial \theta_{s}\partial \theta_{t}\partial \theta_{l}}$ the elements of the third derivative of the log-likelihood, with s,t,l=1,…,d. Finally, let us define the following two quantities:$$v_{1,1}=3\sum_{i=1}^{d}\sum_{j=1}^{d}\ell_{1ij}^{\left(3\right)}\left(\tilde{\theta }\right)\;\Omega_{ij}+3\sum_{i=1}^{d}\sum_{j=1}^{d}\sum_{k=1}^{d}\ell_{ijk}^{\left(3\right)}\left(\tilde{\theta }\right)\;\Omega_{ij}\Omega_{k1}$$
and$$\begin{array}{ccc} v_{3,111} & = & \ell_{111}^{\left(3\right)}\left(\tilde{\theta }\right)+3\sum_{i=1}^{d}\ell_{11i}^{\left(3\right)}\left(\tilde{\theta }\right)\;\Omega_{i1}+3\sum_{i=1}^{d}\sum_{j=1}^{d}\ell_{1ij}^{\left(3\right)}\left(\tilde{\theta }\right)\;\Omega_{ij}\Omega_{j1}\\ & + & \sum_{i=1}^{d}\sum_{j=1}^{d}\sum_{k=1}^{d}\ell_{ijk}^{\left(3\right)}\left(\tilde{\theta }\right)\;\Omega_{ij}\Omega_{k1}\Omega_{11}. \end{array}$$Then, following formula (23) in [5], the SKS approximation of the marginal posterior density $\pi_{m}\left(\psi |y\right)$ can be expressed as(25)$$\pi_{mSKS}\left(\psi |y\right)\propto 2\;\phi \left(h_{\psi };0,\Omega_{11}\right)\;\Phi \left(\alpha_{\psi }\left(h_{\psi }\right)\right),$$
where $h_{\psi }=\sqrt{n}\left(\psi -\tilde{\psi }\right)$ is the rescaled parameter of interest, $\phi (\cdot ;0,\Omega_{11})$ is the density of a Gaussian distribution with mean 0 and variance $\Omega_{11}$, and the skewness component $\alpha_{\psi }\left(h_{\psi }\right)$ is defined as$$\alpha_{\psi }\left(h_{\psi }\right)=\frac{\sqrt{2\pi }}{12n^{3/2}}\left(v_{1,1}h_{\psi }+v_{3,111}h_{\psi }^{3}\right).$$Using (Equation 25), we can derive the SKS tail area approximation of (Equation 9), given by$$P_{mSKS}\left(\psi \geq \psi_{0}|y\right)=\int_{h_{\psi 0}}^{\infty }2\;\phi \left(h_{\psi };0,\Omega_{11}\right)\;\Phi \left(\alpha_{\psi }\left(h_{\psi }\right)\right)\;dh_{\psi },$$
where $h_{\psi 0}=\sqrt{n}\left(\psi_{0}-\tilde{\psi }\right)$. Finally, the marginal SKS approximation of the BDM is given by(26)$$\begin{array}{c} \delta_{H}^{mSKS}=1-2\min\left\{P_{mSKS}\left(\psi \geq \psi_{0}|y\right),1-P_{mSKS}\left(\psi \geq \psi_{0}|y\right)\right\}. \end{array}$$The marginal SKS tail area approximation $P_{mSKS}\left(\psi \geq \psi_{0}|y\right)$, and, thus, also $\delta_{H}^{mSKS}$, can be derived numerically.

### 4.3. Multidimensional Parameters

While the SKS approximation is theoretically elegant, similar to the higher-order modification of the log-likelihood ratio $W^{*}\left(\theta \right)$, it has two main drawbacks. The first one is that it relies only on local information around the mode. The second is that it is computationally intensive because it relies on third-order derivatives (i.e., a tensor of derivatives) of the log-likelihood. The size of this derivative tensor increases cubically with the number of parameters, leading to substantial memory and computational demands, particularly in models with many parameters. Furthermore, quantities such as the moments, marginal distributions, and quantiles of the SKS approximation are not available in closed form, even in the scalar case.

To address these challenges, [6] proposed a class of approximations based on the standard skew-normal (SN) distribution. Their method matches posterior derivatives, aiming to preserve the ability to model skewness while employing more computationally tractable structures. It utilizes local information around the MAP by matching the mode *m*, the negative Hessian at the mode, i.e., $j(\tilde{\theta })$, and the third-order unmixed derivatives vector $t\in R^{d}$ of the log-posterior. Moreover, modern tools for automatic differentiation can greatly facilitate the computation of such higher-order derivatives without manual derivation. The goal is to find the parameters of the multivariate SN distribution ${\mathrm{SN}}_{d}\left(\xi ,\Omega ,\alpha \right)$ that best match these quantities. The notation ${\mathrm{SN}}_{d}\left(\xi ,\Omega ,\alpha \right)$ indicates a *d*-dimensional SN distribution (see e.g., [24], and references therein), with location parameter ξ, scale matrix Ω, and shape parameter α. The matching equations are given by$$\begin{array}{cc} 0 & =-\Omega^{-1}\left(m-\xi \right)+\zeta_{1}\left(\kappa \right)\alpha ,\\ j(\tilde{\theta }) & =\Omega^{-1}-\zeta_{2}\left(\kappa \right)\alpha \alpha^{⊤},\\ t & =\zeta_{3}\left(\kappa \right)\alpha^{\circ 3},\\ \kappa & =\alpha^{⊤}\left(m-\xi \right), \end{array}$$
where $\zeta_{k}\left(\kappa \right)$ denotes the *k*-th derivative of logΦ(κ), and ∘3 represents the Hadamard (element-wise) product. The solution proceeds by reducing the system to a one-dimensional root-finding problem in κ, after which α, Ω, and ξ can be obtained analytically. Ultimately, the marginal distributions are available in closed form as well. Given its tractability, we adopt the derivative matching approach proposed by [6] to derive SKS approximations for models with multidimensional parameters. For the SN model, we can instead easily define the multivariate quantiles.

As suggested in [8,9], an effective approach to defining quantiles in the multidimensional case is to identify the optimal transport (OT) map between the spherical uniform distribution and the target multivariate SN distribution. Considering the inherent relationship between the standard multivariate Gaussian distribution and the spherical uniform distribution, we explore the OT map linking a multivariate SN distribution to a multivariate standard normal distribution. Indeed, given a multivariate standard normal *S* in $R^{d}$, it is well known that U=S/∥S∥ is uniformly distributed on the sphere of radius $\sqrt{d}$ in $R^{d}$. Furthermore, 2(Φ(∥S∥)−0.5) is uniform in (0,1). Thus, the OT map and the quantiles of the multivariate standard Gaussian are coherently defined as a bijection of the norm of the multivariate standard normal vector *S* (the distance from the origin). In particular, we utilize the canonical multivariate SN distribution, derived from applying a rotational transformation, and we consider a component-wise transformation using the univariate SN distribution function and the standard normal quantile function, which delineates a transport map represented as the gradient of a convex function.

From $X∼{\mathrm{SN}}_{d}\left(\xi ,\Omega ,\alpha \right)$, let $\delta =\Omega (\alpha /\sqrt{1+\alpha^{⊤}\Omega \alpha })$. We define a rotation $T_{1}\left(X\right)=QX$ by means of the matrix $Q\in R^{d\times d}$ such that
$Z=Q^{⊤}\left(X-\xi \right)$ aligns the skewness with the first coordinate;in the rotated space, $Z_{1}∼{\mathrm{SN}}_{1}(0,\omega^{2},∥\alpha ∥)$, with $\omega^{2}={\left[Q^{⊤}\Omega Q\right]}_{1,1}$, and $Z_{2:d}$ are Gaussian.
The matrix *Q* is obtained by applying a (rectangular) QR decomposition to the α vector. The vector of means is $E\left(Z\right)=Q^{⊤}\delta \sqrt{2/\pi }$ and the covariance matrix is $V=Q^{⊤}\left(\Omega -\frac{2}{\pi }{\left(Q^{⊤}\delta \right)}^{⊤}Q^{⊤}\delta \right)$. Moreover, the scale parameter of $Z_{1}$ is $\sigma =\sqrt{Q⊤\Omega Q}$ and we denote its mean and variance as $\mu_{1}=E\left[Z_{1}\right]$ and $V_{1}=Var\left(Z_{1}\right)$, respectively.

We define the transport map $T_{2}\left(X\right)$ in the rotated space as$$T_{2}\left(X\right)=\left[\begin{array}{c} \Phi^{-1}\left(F_{\mathrm{SN}}\left(X_{1},0,\sigma^{2},Q^{⊤}\alpha \right),\mu_{1},V_{1}\right)\\ X_{2}\\ \vdots \\ X_{d} \end{array}\right],$$
where $F_{\mathrm{SN}}\left(\cdot \right)$ is the univariate SN cumulative distribution function and $\Phi^{-1}\left(\cdot \right)$ is the standard normal quantile function. In practice, we transform the first component using the univariate SN cumulative distribution function ($F_{SN}$) and the standard normal quantile function ($\Phi^{-1}$) to remove its skewness, while leaving other components unchanged. Note that the SN distribution is closed under linear transformations. In particular, after the rotation, the skewness of the variable *Z* becomes $Q^{⊤}\alpha$ (see [24]). The variable $Z^{\prime }=T_{2}\left(Z\right)$ is now approximately multivariate normal. Finally, we apply an affine transformation to standardize the result. More precisely, we consider $T_{3}\left(X\right)=V^{-1/2}\left(X-Q^{⊤}\delta \sqrt{2/\pi }\right)$, and set $U=T_{3}\left(Z^{\prime }\right)$. The resulting *U* is distributed as a standard normal (see Figure 1).

It follows that, using the SN approximation $\pi_{SN}\left(\theta |y\right)$ for the posterior distribution of θ, the SN approximation of the BDM can be expressed as(27)$$\begin{array}{c} \delta_{H}^{SN}=1-\;\mathrm{Pr}(\chi_{d}^{2}\geq ∥T\left(\theta_{0}\right)∥), \end{array}$$
where $T\left(x\right)=T_{3}\circ T_{2}\circ T_{1}$. The map $T\left(x\right)=T_{3}⊙T_{2}⊙T_{1}$ is the OT map as it is the gradient of a convex function. In particular, $T_{1}$ and $T_{3}$ are affine transformations, and the function $\Phi^{-1}\left(F_{\mathrm{SN}}\left(z,\xi ,\omega ,\alpha \right)\right)$ is monotonically increasing in *z*, hence its integral is convex. Defining$$g\left(Z\right)=\int_{0}^{Z_{1}}\Phi^{-1}\left(F_{\mathrm{SN}}\left(t,\xi ,\omega ,\alpha \right)\right)\;dt+\frac{1}{2}\sum_{i=2}^{d}Z_{i}^{2},$$
then $T_{2}\left(Z\right)=\nabla g\left(Z\right)$. The composite map T(·), used in (Equation 27), is the gradient of a convex function and, thus, it represents the optimal transport map (under quadratic cost) from an SN distribution to a standard normal. The procedure is summarized in Algorithm 1.
**Algorithm 1** Optimal transport from ${\mathrm{SN}}_{d}\left(\xi ,\Omega ,\alpha \right)$ to $N_{d}\left(0,I_{d}\right)$**Input** 
$X\in R^{d}∼{\mathrm{SN}}_{d}\left(\xi ,\Omega ,\alpha \right)$**Output** 
$U∼N_{d}\left(0,I_{d}\right)$Compute $\delta \leftarrow \Omega \alpha /\sqrt{1+\alpha^{⊤}\Omega \alpha }$Compute QR decomposition: Q←QR_decomposition(α)Compute mean and covariance: $\mu_{Z}\leftarrow Q^{⊤}\delta \cdot \sqrt{2/\pi }$, $V\leftarrow Q^{⊤}\left(\Omega -\frac{2}{\pi }\left(Q^{⊤}\delta \right){\left(Q^{⊤}\delta \right)}^{⊤}\right)Q$Compute variance of the rotated first component: $\sigma^{2}\leftarrow {\left[Q^{⊤}\Omega Q\right]}_{1,1}$Set $\alpha_{\mathrm{rot}}\leftarrow Q^{⊤}\alpha$Set $\mu_{1}\leftarrow {\left(\mu_{Z}\right)}_{1}$ and $V_{1}\leftarrow V_{1,1}$Apply rotation: $Z\leftarrow Q^{⊤}\left(X-\xi \right)${$T_{1}$}Compute $u\leftarrow F_{\mathrm{SN}}\left(Z_{1};0,\sigma^{2},\alpha_{\mathrm{rot}}\right)$ Compute $z_{1}^{\prime }\leftarrow \Phi^{-1}\left(u;\mu_{1},V_{1}\right)$Set $Z^{\prime }\leftarrow \left(z_{1}^{\prime },Z_{2},\ldots ,Z_{d}\right)${$T_{2}$}Compute $U\leftarrow V^{-1/2}\left(Z^{\prime }-\mu_{Z}\right)${$T_{3}$}

## 5. Examples of Higher-Order and Skewed Approximations

In the following, we focus on assessing the performance of the higher-order approximations and of the skewed approximations of the BDM in two examples, discussed also in [5,7].

### 5.1. Exponential Model

We revisit Example 1 in [7], where the model for data $y_{1},\ldots ,y_{n}$ is an exponential distribution with scale parameter θ, meaning E(Y)=θ. By employing Jeffrey’s prior, which is $\pi \left(\theta \right)\propto \theta^{-1}$, the resulting posterior distribution is an inverse gamma, characterized by shape and rate parameters equal to *n* and $t_{n}$, respectively, with $t_{n}=\sum_{i}y_{i}$. The quantities for the SKS approximation of the posterior distribution are available in [5] (see Section 3.1), while for the higher-order approximation, we have that q(θ) coincides with the score statistic, i.e., $q\left(\theta \right)=\ell^{\left(1\right)}\left(\theta \right)/i{\left(\theta \right)}^{1/2}$. We analyze how well the two approximations align with the true BDM under the growing sample size (n=6,12,20,40) while keeping the MLE fixed at $\hat{\theta }=1.2$. The MAP values are 1.03 (n=6), 1.11 (n=12), 1.14 (n=20), and 1.17 (n=40).

Figure 2 and Figure 3 and Table 1 report the approximations of the BDM for several candidate values for $\theta_{0}$. In particular, the first-order (IO) approximation (Equation 5), the higher-order (HO) approximation (Equation 13), the SKS approximation (Equation 24), a direct numerical tail area calculation (SKS-num) of (Equation 23) and the SN approximation (Equation 27) are considered. Figure 2 and Figure 3 also display the approximations to the corresponding posterior distributions, where the HO approximation is derived numerically by inverting the tail area. Also, note that the SKS approximation of the posterior distribution is not guaranteed to lie within the interval (0,1), so we practically bounded the BDM in this interval.

The results confirm that the HO and SKS approximations yield remarkable improvements over the first-order counterpart for any *n*. Moreover, they show that the HO approximation of the BDM is almost perfectly over-imposed on the true BDM, especially for values of $\theta_{0}$ far from the MLE. When the value under the null hypothesis is closer to the MLE, the SKS approximation and the numerical tail areas derived from the SKS and SN approximations better approximate the true BDM. Furthermore, the SN approximation more accurately captures the tail behavior of the posterior distribution than the SKS approximation.

### 5.2. Logistic Regression Model

We now consider a real-data application using the Cushing’s dataset (see [5], Section 5.2), which is openly available in the R library MASS. The data were obtained from a medical study on n=27 individuals, aimed at investigating the relationship between Cushing’s syndrome and two steroid metabolites, namely Tetrahydrocortisone and Pregnanetriol.

We define a binary response variable *Y*, which takes the value 1 when the patient is affected by bilateral hyperplasia, and 0 otherwise. The two observed covariates $x_{1}$ and $x_{2}$ are two dummy variables representing the presence of the metabolites. We focus on the most popular regression model for binary data, namely, logistic regression with mean function ${\mathrm{logit}}^{-1}\left(\beta_{0}+\beta_{1}x_{1}+\beta_{2}x_{2}\right)$. As in [5], Bayesian inference is carried out by employing independent, weakly informative Gaussian priors N(0, 25) for the coefficients $\beta =(\beta_{0},\beta_{1},\beta_{2})$.

Figure 4 displays the marginal posterior distributions for $\beta_{1}$ and $\beta_{2}$ obtained via MCMC sampling (black curves), along with the first-order, the SKS, and the SN approximations. The MAP values for the two parameters are −0.031 and −0.286, respectively.

We aim to test the two null hypotheses $H_{0}:\beta_{1}=0$ and $H_{0}:\beta_{2}=0$, corresponding to the null effect of the metabolites’ presence in determining Cushing’s syndrome (indicated by red vertical lines in Figure 4). The exact BDM gives the values 0.592 and 0.932, respectively, indicating that the hypothesized value may support the null hypothesis for the first parameter $\beta_{1}$, whereas the second value suggests a weak disagreement with the assumed value for $H_{0}:\beta_{2}=0$. The SKS approximations of the BDM for the considered hypotheses are 0.612 and 0.935, respectively; the SN approximations are 0.584 and 0.870, respectively; the first-order approximations are 0.512 and 0.891, respectively; while the higher-order approximations provide 0.611 and 0.998, respectively. Finally, the approximations based on the matching priors are 0.477 and 0.862, respectively. Thus, the skewed approximations (SKS, SN) provide the best results.

For the composite hypothesis $H_{0}:\beta_{1}=\beta_{2}=0$, the ground truth is not available; however, in the presence of low correlation between the components, one can approximate it as the geometric mean of the two marginal measures, which is 0.743. The first-order approximation for the BDM gives 0.300, while the SN approximation gives 0.760, revealing that the value under the null is more extreme (see also Figure 5).

## 6. Concluding Remarks

Although the higher-order and skewed approximations described in this paper are derived from asymptotic considerations, they perform well in moderate or even small sample situations. Moreover, they represent an accurate method for computing posterior quantities and approximating $\delta_{H}$, and they make it quite straightforward to assess the effect of changing priors (see, e.g., [25]). When using objective Bayesian procedures based on strong-matching priors and higher-order asymptotics, there is agreement between Bayesian and frequentist point and interval estimation, as well as in significance measures. This is not true, in general, with the *e*-value, as discussed in [14].

A significant contribution of this work is the extension to multivariate hypotheses. We propose a formal definition of the multivariate BDM based on center-outward optimal transport maps, providing a theoretically sound generalization of the univariate concept. By utilizing either the multivariate normal or multivariate SN approximations of the posterior distribution, we can formulate the multivariate quantiles in a closed form, thereby allowing us to derive the BDM for composite hypotheses. Nonetheless, precisely determining or defining these quantiles on the true posterior is challenging, as the transport map may not be available in a closed form and requires solving a complex optimization problem. However, the SN approximation as well as the derived OT map continue to be manageable in high-dimensional settings, whereas typical OT methods generally do not scale efficiently with increasing dimensions.

As a final remark, the high-order procedures proposed and described are tailored to continuous posterior distributions, and their extension to models with discrete or mixed-type parameters warrants further study. Moreover, although the higher-order and skewed methods, alongside SN-based OT maps, offer a useful means for approximating the posterior distributions and computing tail areas, their application might fail in handling complex or irregular posterior landscapes. In such cases, employing integrated computational procedures to find the transport map [26] and utilizing the direct definition of the multivariate BDM could be more appropriate. Furthermore, the utility of the higher-order and skew-normal approximation techniques developed here is not restricted to the Bayesian discrepancy measure. These methods hold considerable promise for approximating other Bayesian measures of evidence. For example, applying these procedures to quantities like the e-value is a natural and compelling direction for future work.

## Figures and Tables

**Figure 1 entropy-27-00657-f001:**
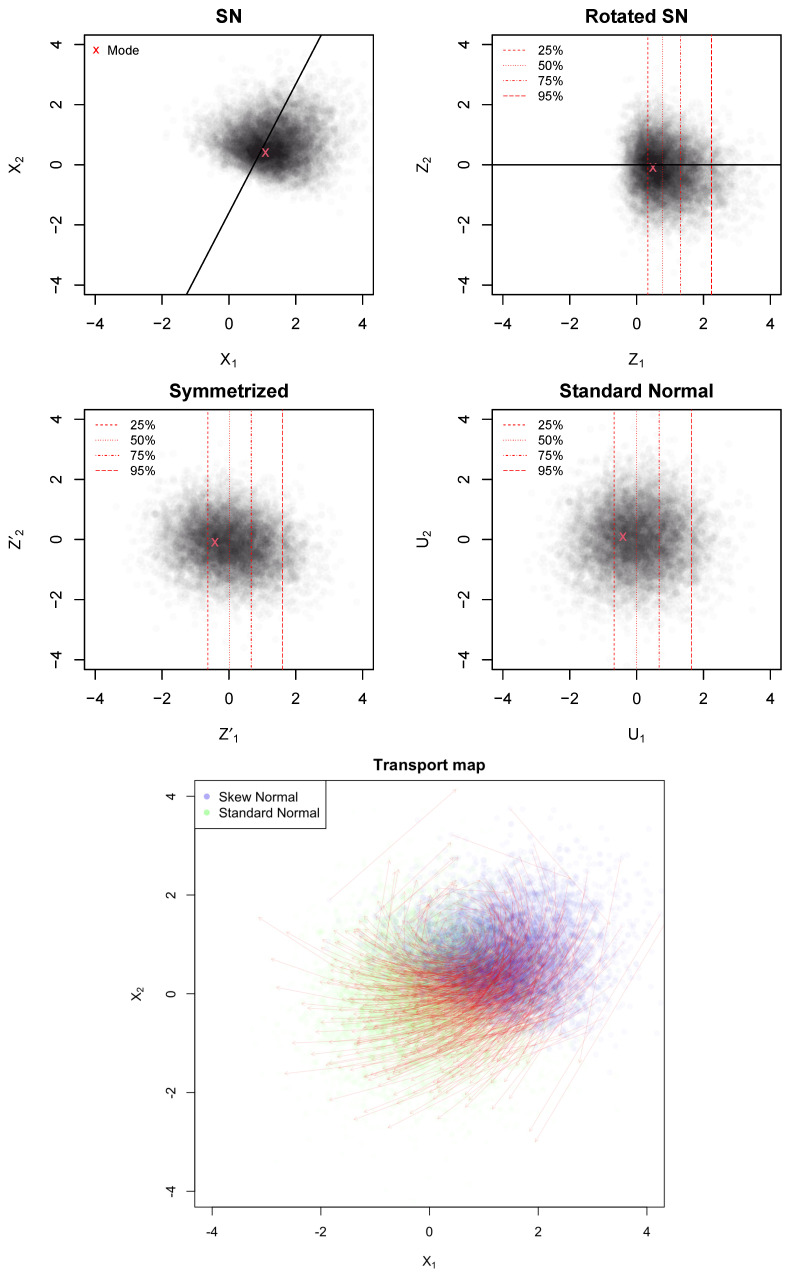
*First panel*: Original SN approximation of a bivariate posterior distribution, with the mode in red and skewness direction indicated by the black line. *Second panel*: Rotated SN distribution aligning the skewness with the first coordinate; red dashed lines show quantiles of the first rotated component. *Third panel*: Symmetrized distribution after applying a univariate marginal transformation. *Fourth panel*: Final standardized and centered normal distribution. *Bottom panel*: Visualization of the optimal transport (OT) map (red arrows).

**Figure 2 entropy-27-00657-f002:**
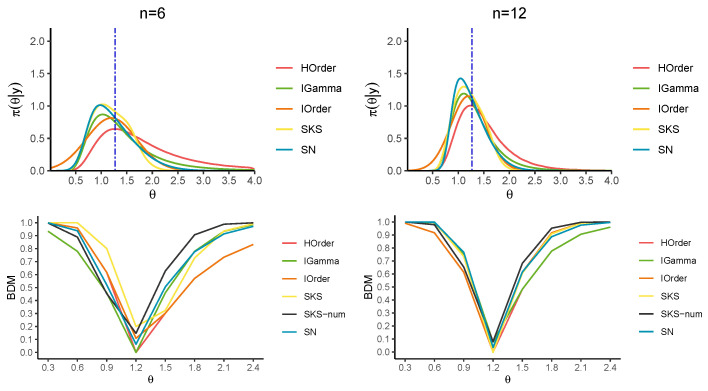
Exact posterior (in green) and approximate posteriors for n=6,12 in the exponential model (top panels). The blue vertical line indicates the posterior median. BDM for a series of parameters (lower panels).

**Figure 3 entropy-27-00657-f003:**
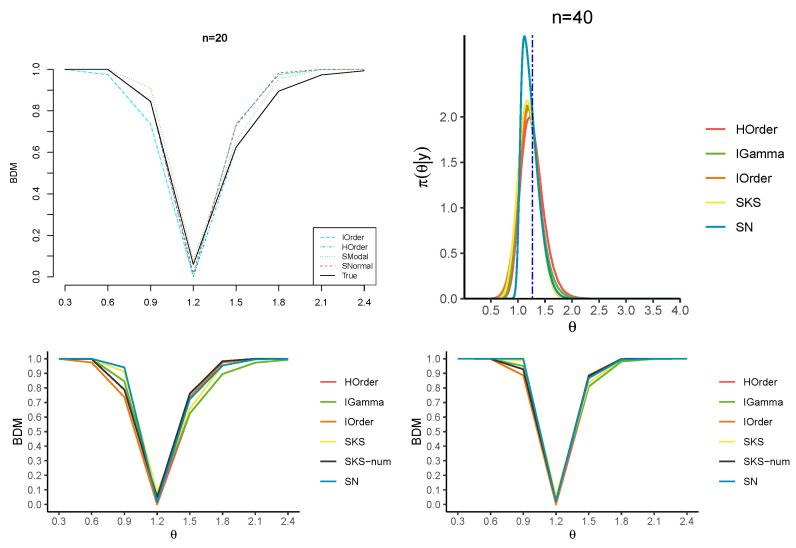
Exact posterior (in green) and approximate posteriors for n=20,40 in the exponential model (top panels). The blue vertical line indicates the posterior median. BDM for a series of parameters (lower panels).

**Figure 4 entropy-27-00657-f004:**
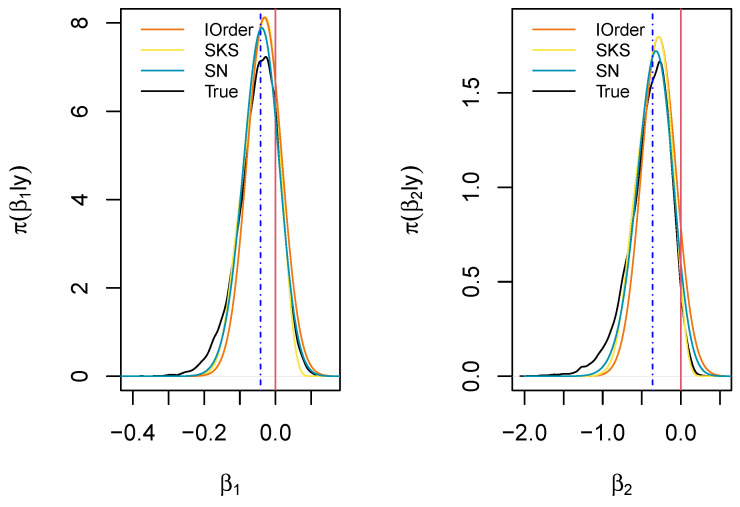
Marginal posterior distributions for the regression parameters of the logistic regression example. The marginal medians are indicated in blue, while the parameters under the null hypothesis are indicated in red.

**Figure 5 entropy-27-00657-f005:**
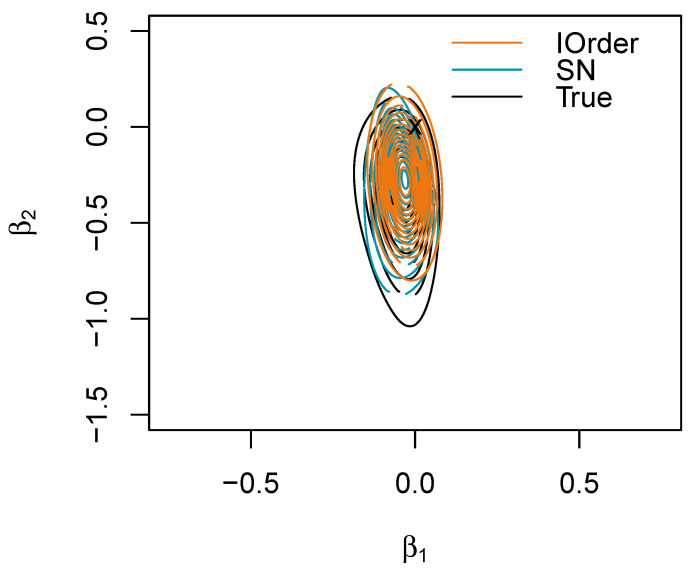
Joint posterior for $\left(\beta_{1},\beta_{2}\right)$ in the logistic regression example with the first-order (IOrder) and skew-normal (SN) approximations. The point (0,0) is marked with a cross.

**Table 1 entropy-27-00657-t001:** BDM for a series of values $\theta_{0}$ for the parameter and increasing sample sizes in the exponential example. The values of the true BDM and the best approximation(s) in each configuration are highlighted in bold.

	$\theta_{0}$	0.3	0.6	0.9	1.2	1.5	1.8	2.1	2.4
n=6	IO	0.93	0.78	0.46	0.00	0.46	0.78	0.93	0.99
	HO	**1.00**	**0.96**	**0.62**	0.00	**0.30**	**0.57**	**0.73**	**0.83**
	SKS	**1.00**	1.00	0.80	0.20	0.32	0.73	0.94	0.99
	SKS-num	**1.00**	0.94	0.53	**0.07**	0.58	0.91	0.99	1.00
	SN	**1.00**	0.94	0.52	0.06	0.51	0.78	0.91	0.97
	**BDM**	**1.00**	**0.96**	**0.62**	**0.11**	**0.30**	**0.57**	**0.73**	**0.83**
n=12	IO	0.99	0.92	0.61	0.00	0.61	0.92	0.99	1.00
	HO	**1.00**	**0.99**	**0.75**	0.00	**0.48**	**0.78**	**0.91**	**0.96**
	SKS	**1.00**	1.00	0.74	−0.00	0.61	0.91	0.99	1.00
	SKS-num	**1.00**	0.99	0.72	0.01	0.64	0.95	1.00	1.00
	SN	**1.00**	1.00	0.77	**0.04**	0.62	0.89	0.98	1.00
	**BDM**	**1.00**	**0.99**	**0.75**	**0.08**	**0.48**	**0.78**	**0.91**	**0.96**
n=20	IO	1.00	0.97	0.74	0.00	0.74	0.97	1.00	1.00
	HO	1.00	**1.00**	**0.85**	0.00	**0.62**	**0.90**	**0.97**	**0.99**
	SKS	1.00	**1.00**	0.91	**0.08**	0.66	0.96	1.00	1.00
	SKS-num	1.00	**1.00**	0.84	0.02	0.73	0.98	1.00	1.00
	SN	1.00	**1.00**	0.94	0.02	0.72	0.95	1.00	1.00
	**BDM**	**1.00**	**1.00**	**0.85**	**0.06**	**0.62**	**0.90**	**0.97**	**0.99**
n=40	IO	1.00	1.00	0.89	0.00	0.89	1.00	1.00	1.00
	HO	1.00	1.00	**0.95**	0.00	**0.81**	**0.98**	1.00	1.00
	SKS	1.00	1.00	0.99	**0.05**	0.83	1.00	1.00	1.00
	SKS-num	1.00	1.00	0.96	0.03	0.87	1.00	1.00	1.00
	SN	1.00	1.00	1.00	0.02	0.87	0.99	1.00	1.00
	**BDM**	**1.00**	**1.00**	**0.95**	**0.04**	**0.81**	**0.98**	**1.00**	**1.00**

## Data Availability

The original contributions presented in this study are included in the article. Further inquiries can be directed to the corresponding author.

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
