# Peer review of "Bayesian Discrepancy Measure: Higher-Order and Skewed Approximations"

_entropy, 2025, doi:10.3390/e27070657_

Round 1
Reviewer 1 Report
Comments and Suggestions for Authors
The research group of Prof. Laura Ventura is characterized by its highly technical but also innovative and interesting work. This research group has been developing original applications of asymptotic analysis to theoretical and applied statistics, and the present paper is yet another excellent work in this line of research.
This paper further develops a Bayesian Discrepancy Measure in multivariate parametric statistical models developed in references [3,4]. This discrepancy measure is able to take advantage of optimal transport mapping techniques to compute highly accurate approximations of statistical quantities of interest.
The paper is well-written, interesting, and understandable, providing some nice application examples. The applications of the asymptotic expansion methods developed in this article follow the path set by references [3,4]. Further opportunities for application of the aforementioned asymptotic expansion methods can be found in the path set by references [11,12,13], requiring nevertheless the development of higher order approximations to integrals of the posterior approximations at an HSD (Highest Surprise Density) region. Such a further development would constitute a nice opportunity for further research for forthcoming articles.
The present article is very good the way it is, constituting a sound piece of mathematical and statistical research, that is ready for publication at Entropy or any other of the best Journals of theoretical Statistics.
Reviewer 2 Report
Comments and Suggestions for Authors
The paper addresses an important gap in Bayesian hypothesis testing by proposing higher-order and skewed approximations for the Bayesian Discrepancy Measure (BDM). The integration of third-order expansions and skew-normal (SN) approximations offers a significant advancement over first-order Gaussian methods.
The extension to multivariate settings using Optimal Transport (OT) maps is innovative and aligns with modern statistical methodologies
Weaknesses and Areas for Improvement
1- While the SN approximation is tractable, its reliance on third-order derivatives (e.g., (v{1,1}, v{3,111}) becomes computationally prohibitive in high-dimensional models. The paper should discuss potential workarounds (e.g., sparse tensor approximations).
2-The OT map construction (Section 4.3) is theoretically elegant but lacks practical guidance. A step-by-step algorithm or pseudocode would enhance reproducibility.
3-The definition δH=∥FP±(θ0)∥ (Equation 11) is abstract. A geometric interpretation (e.g., distance from the posterior centroid) would improve accessibility.
4-The impact of non-matching priors on higher-order approximations (Section 3.2.1) is underexplored. A sensitivity analysis (e.g., using weakly informative vs. subjective priors) would strengthen robustness claims.
5-Symbols like r∗(θ), q(θ), and W∗(θ) are reused across sections, risking confusion. A notation glossary or appendix would help readers.
6-Figures 2–4 lack axis labels and legends in some panels, reducing clarity. Standardizing formatting and adding captions with explicit takeaways is advised.
7-The paper mentions the likelihoodAsy R package but does not provide code or data for replicating examples. Sharing code (e.g., GitHub link) would enhance transparency.
8-For small sample sizes (e.g., n=6), the SKS approximation produces values outside [0,1] (Table 1). This should be addressed with truncation or reparameterization.
9-The paper focuses on comparisons to first-order methods but does not benchmark against other Bayesian measures (e.g., e-value, FBST). A table contrasting properties (e.g., calibration, invariance) would contextualize contributions.
10-Citations are heavily weighted toward pre-2020 literature. Incorporating 2023–2024 advances (e.g., neural transport methods [9]) would modernize the discussion.
Recommendations for Revision
- Add pseudocode for implementing OT maps and SN approximations.
-
Discuss computational shortcuts for high-dimensional third-order derivatives (e.g., automatic differentiation).
- Include a sensitivity analysis for prior choice in higher-order approximations.
-
Benchmark BDM against the e-value or FBST in the examples.
- Standardize notation and provide a glossary.
-
Improve figure readability with labeled axes, legends, and captions.
- Publish code/data in a public repository (e.g., Zenodo, GitHub).
-
Clarify numerical adjustments (e.g., bounding δHSKS in [0,1]).
- Expand the Related Work section to include recent Bayesian nonparametric methods.
-
Address the Jeffreys-Lindley paradox in greater depth, given its relevance to precise hypothesis testing.
Reviewer 3 Report
Comments and Suggestions for Authors
This manuscript is based on the authors' earlier work on Bayesian discrepancy measure, a method proposed for hypothesis testing under the Bayesian framework. It leverages classical frequentist tail probability and Bayesian approximation tools to improve BDM beyond first-order approximation by developing higher-order and skewness approximations and extension to multivariate setting. This work offers valuable insights to match Bayesian and frequentist inference, especially under non-Gaussian posterior distributions.
There are a few comments and suggestions I would like to make about the manuscirpt:
- The first-order approximation of BDM seems to be a direct application of the Bernstein–von Mises theorem. The author may consider clarifying this connection early in the paper.
- In page 6, can the authors clarify for the formula of r(\theta), is it a likelihood root of l(\theta) or posterior \pi(\theta|y)?
- Could the author elaborate what is the difference between the procedure in Section 3.2 and the procedure corresponds to formula (2.17) in Reid, N. (2003). Asymptotics and the theory of inference. The Annals of Statistics, 31(6), 1695-2095.
- Similarly, the procedures in Section 4, at least the one in Section 4.1, is just simple application of the results of Durante, D., Pozza, F., & Szabo, B. (2024). Skewed Bernstein–von Mises theorem and skew-modal approximations. The Annals of Statistics, 52(6), 2714-2737. Could the author clarify any innovation?
- For the heavy tailed case, could the authors try t-distribution with degrees of freedom less than 4 for the simulations?
